# Vessel Anatomical Features of ‘Picual’ and ‘Frantoio’, Two Olive Cultivars Different in Resistance against Verticillium Wilt of Olive

**DOI:** 10.3390/plants12162910

**Published:** 2023-08-10

**Authors:** Antonio Santos-Rufo, Martín Molina-Molina, Esteban Alcántara-Vara, Carlos Weiland-Ardáiz, Fco. Javier López-Escudero

**Affiliations:** 1Excellence Unit ‘María de Maeztu’ 2020-23, Department of Agronomy, Campus de Rabanales, University of Cordoba, 14071 Cordoba, Spain; g12momom@uco.es (M.M.-M.); ag1alvae@uco.es (E.A.-V.); ag2loesj@uco.es (F.J.L.-E.); 2Department of Agroforestry Sciences, ETSI University of Huelva, 21007 Huelva, Spain; weiland@dcaf.uhu.es

**Keywords:** anatomy, ‘Frantoio’, *Olea europaea*, ‘Picual’, *Verticillium dahliae*, verticillium wilt of olive, xylem vessels

## Abstract

The olive tree (*Olea europaea*), a non-tropical woody crop that occupies the largest area in the world, is severely affected by the fungus *Verticillium dahliae* worldwide. In this regard, there is currently detailed information on the level of resistance to this pathogen in the main olive varieties. However, there is little information on quantitative aspects of its anatomy and on the existence of anatomical differences between varieties that could be related to the differential resistance response observed. In the present work, a quantitative study of the xylem of ‘Picual’, susceptible, and ‘Frantoio’, resistant, to *V. dahliae* is carried out. This study also provides quantitative data on the xylem in different areas of the plant, an aspect on which there is not much information for the olive tree. Among the parameters evaluated, it is probably the greater conductive capacity in the xylem tissue that ‘Frantoio’ presents, mainly due to the greater density of its vessels, which has a more relevant role in the resistance and natural recovery that this cultivar manifests to the disease. In any case, these constitutive anatomical differences, and those others that can be induced in plants during infections, should be investigated in future work that includes inoculation with the pathogen.

## 1. Introduction

Verticillium wilts or vascular wilts caused by *Verticillium dahliae* constitute one of the most important and destructive groups of plant diseases in most temperate climate zones around the world, although they have also been described in tropical and subtropical areas [1]. Olive verticillium wilt is considered the most serious disease of this crop in most countries where this crop is of strategic economic importance [2]. The causes are the severity of its attacks, its distribution, and the ease of dispersal of the pathogen at short and long distances, the economic importance of the damage it causes, and the difficulties presented by its control.

The use of resistant varieties is probably the most economical, effective, and environmentally friendly control method. However, in vascular diseases, complete resistance or absence of infection does not exist or is rare [1,3].

The defensive or resistance mechanisms can be structural or biochemical. The structural ones act as a physical barrier, limiting the penetration of the pathogen and its spread inside the plant. Biochemical mechanisms are based on the production of substances that are toxic to the pathogen or that create unfavorable conditions for its growth. Both types of mechanisms can be constitutive and inducible [1].

In tracheomycoses, for the disease to develop, the pathogen must access the vascular system of the plant and colonize the xylem extensively and intensely [4]. In this sense, the resistance responses of the plant are expressed as follows: (a) resistance to the penetration of the pathogen into the vascular system or extravascular resistance; and (b) resistance to vascular colonization once the pathogen has gained access to the xylem [1,5,6].

Once the pathogen accesses the vascular system, the longitudinal colonization of the plant depends fundamentally on the conidia, while mycelial growth is essential for the transverse distribution of the pathogen. In this way, the plant can be systemically colonized unless its defensive mechanisms can contain, protect, or delay the invasion by the pathogen [6]. The reduction of transport of conidia in the xylem constitutes the first resistance requirement [7]. In resistant cultivars, the delayed development of the disease is associated with a restriction and greater discontinuity in vascular colonization and a lower percentage of laterally invaded vessels [8].

Resistance mechanisms in vascular tissues can be divided into physical and biochemical mechanisms. In this context, differences in the organization of the vascular system may have an important effect on resistance against vascular pathogens. For example, the variations in susceptibility found among cultivars of elm (*Ulmus* spp.) to *Ophiostoma novo-ulmi* have been attributed to differences in length, diameter, and grouping of the vessels, as well as to the distribution of the xylem parenchyma [5,9]. In other pathosystems such as alfalfa/*Verticillium albo-atrum*, it has also been suggested that certain characteristics of the xylem vessels, such as size, number, or distribution, may influence host resistance, limiting the longitudinal movement of conidia through the pathosystem xylem, or the transverse colonization of new xylem vessels [10].

In olive trees, where a polygenic horizontal resistance operates, the resistance of hundreds of varieties to olive Verticillium wilt has been evaluated [2]. The reference cultivars in all controlled environment, semi-controlled, or field experiments have been ‘Picual’ as susceptible and ‘Frantoio’ as resistant. However, there are few studies that delve into the resistance mechanisms in these varieties. In other species affected by vascular pathogens, some data are available. For example, in the *Ulmus* spp./*Ophiostoma novo-ulmi*, anatomical studies of the xylem have been carried out in species with different susceptibility. The least susceptible species, *Ulmus pumila*, had a smaller mean vessel diameter, a lower proportion of large vessels in earlywood, and a lower theoretical hydraulic conductance, compared to the more susceptible *Ulmus* minor species [11,12]. It is thought that the characteristics of *Ulmus pumila* contribute to less risk of cavitation, to limit the spread of the pathogen through the vascular system, and to facilitate the compartmentalization of the disease. In addition, individuals with smaller diameter vessels can form tyloses more quickly and in greater numbers, making it difficult for the pathogen to spread [13,14].

In relation to olive Verticillium wilt, there is detailed information on the level of resistance to *V. dahliae* of the main varieties [15,16,17,18]. However, there is little information on quantitative aspects of its anatomy and on the existence of anatomical differences between varieties that could be related to the differential resistance response observed.

Therefore, the aims of this work were to provide quantitative data regarding the anatomy of the xylem in different areas of the planting olive seedlings grown under controlled conditions and establish a comparison of these characteristics between two olive cultivars, with different levels of resistance against *V. dahliae*.

## 2. Results

### 2.1. Anatomical Study of the Basal Zones of the Main Stems of Olive Plants

The average area of the cross section evaluated was 27 mm^2^ in ‘Picual’ and 41 mm^2^ in ‘Frantoio’. The percentages of surface occupied by the different tissues were similar in the two varieties, with ‘Picual’ presenting somewhat more xylem and less phloem than ‘Frantoio’ (Figure 1). The highest percentage of average surface in both varieties is occupied by the xylem (54–59%), followed by the phloem (19–24%).

**Note that** mm had been marked for the density of rays and a rectangular sector with an approximate area of 0.56 mm^2^ for the density of vessels and the average area of the vessel. The total number of vessels measured, counting the three repetitions, was 825 in ‘Picual’ and 1255 in ‘Frantoio’. The results (Table 1) show that ‘Frantoio’ had significantly higher values of vessel density, percentage of conductive cross-sectional area (TCS), and relative theoretical conductance (RTC). There were no significant differences between varieties in ray density and average vessel surface.

In Figure 2, the distribution of vessels is represented, grouped according to surface intervals. In both varieties, approximately 80% of the vessels had a surface between 200 and 800 μm^2^, with vessels larger than 1000 μm^2^ being observed. ‘Frantoio’ has a lower percentage of vessels with a smaller surface and a higher percentage of vessels with a larger surface. Thus, approximately 31% of the vessels had a surface area less than 400 μm^2^ and 69% had a surface area greater than 400 μm^2^, while for ‘Picual’ it was 44% and 56% for those same intervals.

### 2.2. Anatomical Study of the Apical Zones of the Main Stems of Olive Plants

The average area of the cross section evaluated was 13 mm^2^ in ‘Picual’ and 14 mm^2^ in ‘Frantoio’. The percentages of surface occupied by the different tissues (Figure 3) were very similar in both varieties, with ‘Picual’ presenting a little more pith than ‘Frantoio’. The highest percentage of surface is occupied by the xylem (45%), followed by the phloem (22%).

To determine the quantitative characteristics of the xylem, an arc length and a rectangular sector area with dimensions like the previous section were considered. The total number of vessels measured, considering the three repetitions, was 1090 in ‘Picual’ and 1458 in ‘Frantoio’. The results (Table 2) show that ‘Frantoio’ had significantly higher values of vessel density and TCS. In ray density, average vessel surface area, and RTC, there were no significant differences between varieties.

In Figure 4, the distribution of vessels is represented and grouped according to surface intervals. In ‘Picual’, approximately 93% of the vessels had a surface between 1 and 600 μm^2^, while in ‘Frantoio’, approximately 78% of the vessels had a surface between 200 and 800 μm^2^, and in ‘Frantoio’ vessels with more than 1000 μm^2^ were even observed. ‘Frantoio’ has a lower percentage of vessels with a smaller surface and a higher percentage of vessels with a larger surface. Thus, approximately 45% of the vessels had a surface area less than 400 μm^2^ and 55% had a surface area greater than 400 μm^2^, while for ‘Picual’ it was 68% and 32% for those same intervals.

In Figure 5 photographs of cross sections are shown, in which different tissues and elements of the xylem are identified.

### 2.3. Anatomical Study of the Side Shoots of Olive Plants

The average area of the cross section evaluated was approximately 3 mm^2^ in both varieties. The percentage of surface occupied by the phloem (Figure 6) was very similar for the two varieties. ‘Picual’ presented a higher percentage of pith and cortex than ‘Frantoio’. The xylem and sub-epidermis percentages were higher for ‘Frantoio’. The highest percentage of surface is occupied by the cortex (25–30%), followed by the xylem (20–25%).

To determine the quantitative characteristics of the xylem, an arc with an approximate length of 1.1 mm was marked for the density of rays, and a rectangular sector with an approximate area of 0.25 mm^2^ was marked for the density of vessels and the average surface area of the vessel. The total number of vessels measured, counting the three repetitions, was 872 in ‘Picual’ and 1341 in ‘Frantoio’. The results (Table 3) show that ‘Frantoio’ had significantly higher values of TCS and RTC. In ray density, in vessel density, and in mean vessel surface, there were no significant differences between varieties.

In Figure 7, the distribution of vessels is represented, grouped according to surface intervals. In ‘Picual’, approximately 85% of the vessels had a surface between 50 and 200 μm^2^, while in ‘Frantoio’, approximately 68% of the vessels had a surface between 100 and 250 μm^2^, with some vessels up to 400 μm^2^. ‘Frantoio’ had a lower percentage of vessels with a smaller surface and a higher percentage of vessels with a larger surface. Thus, approximately 14% of the vessels had a surface area less than 100 μm^2^ and 86% of the vessels has a surface area greater than 100 μm^2^, while for ‘Picual’ it was 44% and 56% for those same intervals.

In Figure 8, photographs of cross sections are shown, in which different tissues and elements of the xylem are identified.

### 2.4. Anatomical Study of the Leaf Petioles of Olive Plants

The average area of the cross section evaluated was 1.1 mm^2^ in the two varieties. The percentages of surface occupied by the different tissues (Figure 9) were similar in both varieties, with ‘Frantoio’ presenting slightly more cortex and less epidermis than ‘Picual’. The highest percentage of surface is occupied by the cortex (63–68%), followed by the epidermis (15–20%).

To determine the quantitative characteristics of the xylem, an arc of approximately 0.5 mm was marked for the density of rays, and an approximate surface of 0.07 mm^2^ was marked for the density of vessels and the average surface of the vessel. The total number of vessels measured, considering the three repetitions, was 657 in ‘Picual’ and 1170 in ‘Frantoio’. The results (Table 4) show that ‘Frantoio’ had significantly higher values of vessel density, mean vessel area, TCS, and RTC. In radio density, there were no significant differences between the varieties.

In Figure 10, the distribution of vessels is represented, grouped according to surface intervals. In ‘Picual’, approximately 98% of the vessels had a surface between 1 and 100 μm^2^, while in ‘Frantoio’, approximately 82 % of the vessels had a surface in the same interval, and in ‘Frantoio’ vessels up to 250 μm^2^ were even observed. Both varieties have a higher percentage of vessels with a smaller surface and a lower percentage of those with a larger surface. Thus, in ‘Frantoio’ approximately 18% of the vessels had a surface greater than 100 μm^2^, while for ‘Picual’ it was 2% in the same interval.

In Figure 11 photographs of cross sections are shown, in which different tissues and elements of the xylem are identified.

## 3. Discussion

The architecture of the xylem plays a fundamental role in vascular diseases, not only because it constitutes the main path of colonization of the pathogen, but also because the damage that occurs comes from the alteration of its water-conducting function, which even causes leaves and branches to wilt and even causes the death of the tree.

There are different characteristics of the xylem that may be related to both aspects, including colonization (longitudinal and transverse) and water transport. Among them, the following can be considered: width and length of the vessels, density and grouping of the vessels, size and abundance of pits, ability to form barriers to colonization, such as tyloses or gums inside the vessels, or as tissues of radial isolation towards the new xylem. The relationship between xylem characteristics and resistance has not been extensively studied, and the results provided cannot be considered general or conclusive. In addition, it must be considered that other biochemical mechanisms operate in resistance.

In the present work, a quantitative study of the xylem of two olive varieties with different levels of resistance to *V. dahliae* is made. The study also provides quantitative data on the xylem in different areas of the plant, an aspect on which not much information is available for the olive tree. The work is carried out on non-inoculated plants, so the possible changes induced by the pathogen, which could also contribute to resistance, are not studied.

In the aerial part, the percentage of xylem in the cross section increased from the petiole (6%), to the bud (20–25%), to the apical zone of the stem (45%), to its basal zone (55–60%). These differences are due to an increase in secondary growth, in which the tissue that is produced in greater quantity is the xylem. For each zone, the surface percentages of the different tissues were similar for the two varieties, indicating a comparable state of maturity.

The characteristics of the xylem varied according to the area, with some differences between varieties. Thus, the density of rays decreased in the areas with more secondary growth, but without differences between varieties (40–41 rays/mm in petiole vs. 21–24 rays/mm in the basal zone of the stem). The conducting capacity of water, estimated through the TCS or the RTC, was higher in the stem than in the shoots and petioles. These differences are a consequence of variations in the surface area and density of the vessels. In the stem, compared to shoots and petioles, the average surface area of the vessels was greater and their density was lesser.

In all zones, the conductive capacity parameters, TCS and RTC, were higher in the ’Frantoio´ variety (the only non-significant difference was for RTC in the apical zone of the stem). This variety presented higher density of vessels than ´Picual´ in all areas (only in bud the difference was not significant). The average surface of the vase was also greater in ´Frantoio´, although with a significant difference only in the petiole. This last trend is reinforced when the distribution of vessels in surface intervals is studied, observing that in general the highest percentages of large vessels correspond to ´Frantoio´.

Variations in vessel width and density within the same plant have been widely described. As an example, in birch trees there are variations according to the position of the branch and in different areas of it [19]. In general, there is an inverse relationship between the density of vessels and the average surface of the vessel, which has been described in different woody species, such as birch [19], avocado [20], oak [21], eucalyptus [22], peach tree [23], and gall oak [21]. The presence of narrower vessels is an advantage in reducing the risk of cavitation, compensating for the lack of conductive capacity with an increase in vessel density [24,25,26,27,28]. This situation is favorable in areas where there is greater tension in the xylem water, such as the areas close to the organs with the most transpiration, specifically the branches and stem closest to the crown or petioles. This situation is also observed in plants subjected to water or saline stress. Thus, for example, boxwood responds to increased aridity by increasing the density of vessels [21], or cotton and sorghum respond to saline stress with an increase of the density of vessels and a decrease in their diameter [29,30].

Trifilò et al. (2007) [31] studied olive shoots of the year in two genotypes of the ‘Leccino’ variety, one vigorous (Minerva) and the other dwarfing (Dwarf). They obtained a vessel density of 405–540 vessels/mm^2^, a mean vessel surface of 175–314 µm^2^, and a conductive cross-sectional surface of 12–14%, thus presenting values that are comparable to those of the shoots in our study (519–588 vessels/ mm^2^, 120–180 µm^2^, and 6–11%, respectively). These authors related the greater vulnerability to cavitation of the Minerva genotype with its greater mean vessel surface area, greater proportion of larger-diameter vessels, and lower vessel density. In petioles, wider vessels have also been found in Minerva, with 19% between 177–314 µm^2^ (maximum surface area found) compared to 8% in Dwarf for the same interval [32]. In our work, ‘Frantoio’ presented 18% of petioles with vessels between 100–250 µm^2^ (maximum surface area found), with this percentage being 2% in ‘Picual’.

The differences between varieties obtained for the xylem of different areas of the aerial part could contribute, together with other mechanisms, to the different resistance responses to the pathogen. In any case, this work has not directly investigated how and to what extent anatomical parameters can influence resistance to verticillium wilt. However, the general starting hypothesis has been that some constitutive anatomical differences could be related to differences in resistance of the two cultivars to infections by the pathogen.

Among the parameters investigated, probably the higher density of vessels in the resistant cultivar may have a relevant role in the resistance shown by these plants during the development of infections. In general lines, although the variations of the anatomical parameters due to the establishment of the infection and the development of the colonization may be more complex, this fact could be summed up in that ‘Frantoio’ presents a greater conductive area in the xylem tissue, mainly due to the increase of the number of vessels, and this contributes to increased resistance to disease.

It must be considered that these observations have been made on non-inoculated control plants, so the results only reflect constitutive differences between both genotypes. It is not ruled out, as has been shown in other pathosystems such as cotton/*Fusarium oxysporum* f. sp. *vasinfectum* [29], that these differences are accentuated during the development of the disease, because of a differential active growth of the xylem of both cultivars, induced by the infection and the colonization of the pathogen.

In fact, the natural recovery from the disease that occurs in rooted cuttings of plants of the resistant genotype, like those used in this study, seven weeks after being artificially inoculated, could correspond to a similar vegetative reaction on the part of the plant [33]. These studies show how, regardless of the level of resistance, various olive cultivars artificially inoculated with the pathogen under controlled conditions optimal for olive growth initially suffer a stoppage of vegetative growth compared to their non-inoculated controls. Approximately 4 weeks after inoculation, the first consistent symptoms of the disease appear on plants of susceptible cultivars such as ‘Picual’, which are only slight, or absent, on resistant genotypes such as ‘Frantoio’. However, 8–9 weeks after inoculation, remission of symptoms begins to be noticed in resistant cultivars, accompanied by new shoot growth, a consequence of the reactivation of the apical meristems, a phenomenon that is not observed in susceptible cultivars. It is probable that together with this longitudinal growth, a reactivation of the vascular cambium that would result in the production of new conductive tissue with an increase in the number of vessels, which if higher in ‘Frantoio’, would facilitate the recovery of the plant. This is undoubtedly one of the objectives that should be addressed in future research.

The influence of variations in the conductive capacity of the xylem on resistance to vascular diseases has been interpreted in different ways in various pathosystems. Although there are few studies in which the values of some of the anatomical parameters involved (vessel density, vessel size, etc.) are related to the response to this type of infection, some examples can illustrate the different hypotheses put forward.

One of them is the work of Turco et al. (2002) [29] in cotton infected with the vascular pathogen *Fusarium oxysporum* f.sp. *vasinfectum*. These authors observed, in two varieties of different susceptibility (‘Acala´, resistant, and ´Cocker´, susceptible), that in conditions of salinity stress there was an increase in the severity of wilting. This fact has also been observed in other herbaceous or woody species [1,30] and in the olive cultivar ‘Picual’ [34]. In the studies on cotton, in the absence of infections, the density of vessels remained similar in both varieties (80/mm^2^), while the increase in salinity and the consequent increase in severity was accompanied by an increase in the density of vessels (200/mm^2^), which was somewhat higher in the susceptible variety. According to these authors, the increase in vessel density under salinity conditions would reduce the effects of cavitation, but would facilitate colonization by the pathogen, thus increasing the severity of the disease. In our studies in olive trees, the opposite has been argued, proposing that an increase in the density of vessels would favor the reduction of the disease since it would help the plant to increase its conductive capacity.

Other hypotheses are raised by a second example provided by Solla and Gil (2002) [9] and carried out in the elm/*Ophiostoma novo-ulmi* pathosystem. These authors studied the relationship between the diameter and the density of vessels in the branches, with the susceptibility of various genotypes of elms to Dutch disease. A greater diameter was found in the vessels of the third growth ring in the genotypes most susceptible to the disease. The authors suggest that this fact, probably induced by the infection and colonization of the pathogen, would cause a greater risk of cavitation and a more rapid and severe development of wilt. In addition, they analyzed the conductive surface resulting from excluding the largest vessels (the elm presents ring porosity), assuming that they are the first to stop being functional due to cavitation [35]. The most resistant elms had a higher density of vessels, excluding the large ones, thus contributing the greater conductive capacity to the increase in resistance to the disease. This hypothesis does coincide with the results of our work, in which ‘Frantoio’ has shown to have a greater conductive capacity than ‘Picual’ by presenting a higher density of vessels. This argument is also included in Beckman (1987) [6], indicating that a beneficial response for plants that have lost conductive capacity is to increase the degree of vascularization, and he cites works that show this response in *Fusarium*-infected tomato plants [36,37] and in hops infected by *Verticillium* [38].

In any case, these hypotheses about the anatomical variations induced in the morphology, number, or distribution of vessels, which as has already been said must be confirmed in pathogenicity experiments, could only partially explain the differences in resistance between cultivars. To these hypotheses, we should add the variations in other anatomical and biochemical parameters, constitutive or induced by infections, related to the longitudinal movement of the pathogen, and to the transverse colonization of other vessels. Deepening these aspects would be of interest to clarify the role of the anatomy of the vascular system in the differences in resistance to verticillium wilt observed between olive cultivars.

## 4. Materials and Methods

### 4.1. The Plant Material, Growth Conditions, and Experimental Design

The plant material used consisted of 9–12-month-old olive seedlings of the ‘Picual’ and ‘Frantoio’ varieties, from the IFAPA World Olive Germplasm Bank (CIFA Alameda del Obispo) in Córdoba, propagated by rooting semi-woody cuttings under mist [39].

These cultivars were chosen for their differential response to *V. dahliae*. ‘Picual’ has been shown to be extremely susceptible to the defoliant pathotype and susceptible to the non-defoliant pathotype, while ‘Frantoio’ is moderately resistant or resistant against the defoliant and non-defoliant isolates, respectively [15,16,17,18,40].

The seedlings were grown in plastic bags of about two liters of capacity, with a sand and lime substrate (1:1) fertilized with a slow-release fertilizer (Osmocote^®^ 18-6-12). The plants grew in a controlled environment chamber, with a temperature of 22 ± 2/20 ± 2 °C day/night, illumination of 216 μmol·m^2^·s¹ of fluorescent light for 14h/day, and relative humidity of 50–70%. The seedlings were irrigated every 2–3 days with 50–100 mL of water as needed, adding every two weeks 150 mL of a complete nutrient solution (Hakaphos green 15-10-15, 2 Mg, with Kifix-Mix microelements, Basf).

The experimental design was totally randomized, using three repetitions for each variety. At the time of sampling, the seedlings were formed by a main axis of an approximate height of 1 m, and several lateral shoots.

### 4.2. Sampling of Plant Material

The vegetative organs under study were main stems, lateral shoots, and petioles. Pruning shears were used to obtain the different samples. The sampled material was homogeneous in size, origin (apical/basal/intermediate), and state of development (vigor). The criteria followed to sample each organ are described below.

Five samples per plant were taken from the root, coming from intermediate zones of lateral roots. Previously, the root ball extracted from the rearing bag was washed with plenty of water to remove any adhering substrate remains. The selected roots came from young areas of new growth and with a good exterior appearance (light tones) [41]. Three cuts were made for each sample, making for 15 cuts per plant. Younger roots with a smaller diameter were also sampled, but the methodology used to obtain cuts did not allow obtaining sufficient quality for their study. 

Two zones were selected from the main stem, one basal and the other apical, making three cuts of each one per plant. The lateral shoots, one per plant, were chosen among those of normal vigor and located at the insertion with the 3–4th basal node of the main stem. The sample was obtained from the 6–7th internode, counted from the apical end of each shoot, making three cuts for each sample.

The sampled petioles, one per plant, were taken from fully expanded leaves of normal vigor, coming from the most developed lateral shoot located at a medium height with respect to the base (3–4th node) and located at the 4–5th node counting from the apical end of the selected shoot. Three cuts per sample were made.

### 4.3. Obtaining Cross Sections

The samples were placed in petri dishes with deionized water (25 mL/plate) until they were processed to avoid drying out. Subsequently, they were included in carrot molds and introduced into the manual microtome (Shibuya). Serial cross sections 20–25 μm thick were obtained with a blade (Feather).

The sections were collected with a brush and placed in a watch glass with deionized water to avoid dehydration and contamination. Subsequently, the sections were mounted on slides (76 × 26 mm) and stained with a few drops of 0.05% Tolouidine blue [42] for 3 min. After this time, excess dye was removed and the sample was hydrated, finally covering it with a coverslip (20 × 20 mm). With this staining, the cellulosic walls acquire a pink-purple hue, while the lignified walls turn blue-green.

The assemblies made were examined under an optical microscope (Nikon, Eclipse 80i, Tokio, Japan) with objectives of different magnifications depending on the samples (×1, ×4, ×10), or with a binocular magnifying glass (Nikon, SMZ-2T, Tokio, Japan) for the larger sections diameters. The microphotographs were obtained using a built-in image capture device (Nikon, Digital Camera Dxm1200C, Tokio, Japan) for subsequent analysis with the NIS-Elements D 2.3 software (Nikon, Japan) with which the different morphometric measurements were made. Once the images were taken, the mounts were sealed with transparent fixative enamel to preserve them.

### 4.4. Anatomical Parameter Evaluation

In the microphotographs, the total area of the section considered plus the areas of the different tissues were delimited and measured, determining the percentage occupied by each of them (Figure 12a,b). To obtain the density of xylem rays (radios/mm), an arc perpendicular to them with a known length was marked. To carry out the measurements of the xylem vessels, a rectangular sector of known surface was marked, counting and measuring the area of all existing vessels in it. In the case of the petioles, the total surface of the xylem was studied (Figure 12c). From these data, the mean vessel surface area (μm^2^), vessel density (vessels/mm^2^) (Figure 12d), conductive cross-sectional area, TCS (%), and relative theoretical conductance, RTC (μm^4^/μm^2^) were obtained. The TCS was calculated by dividing the occupied area by the vessels in a sector, divided by the total area of the sector, and multiplying the result by 100 [43]. The RTC, based on the Hagen-Poiseuille equation [44], was calculated by dividing the sum of the fourth power of the radii of the vessels of a sector by the area of the sector. Likewise, the distribution of vessels according to the surface of their section was obtained, expressing the percentages of vessels in different surface intervals.

### 4.5. Statistical Analysis

With the mean data of each tree, an analysis of variance (ANOVA) was performed, and the means of the two varieties were compared by means of multiple contrasts of equality LSD (Fisher’s Least Significant Protected Difference) with a significance level of 0.05. Analysis of variance was performed using the General AOV/AOCV procedure of the Statistix 10 statistical package (Analytical Software, 2008, Tallahassee, FL, USA).

## Figures and Tables

**Figure 1 plants-12-02910-f001:**
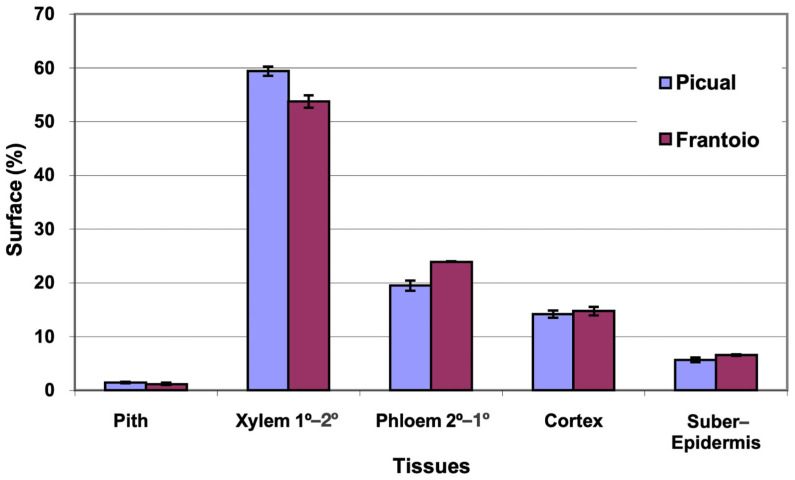
Percentage of the cross section corresponding to different tissues of the basal zones of the main stems of ‘Picual’ and ‘Frantoio’ olive plants. The values represent the mean ± standard error (*n* = 3).

**Figure 2 plants-12-02910-f002:**
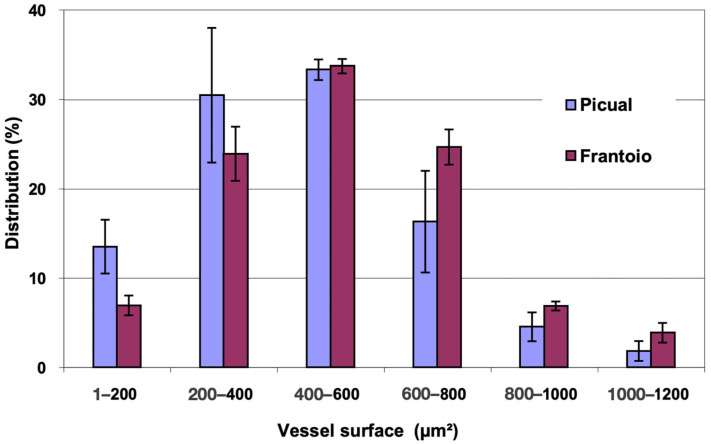
Percentage distribution of the vessels, according to their surface intervals, of the basal zones of the main stems of ‘Picual’ and ‘Frantoio’ olive plants. The values represent the mean ± standard error (*n* = 3).

**Figure 3 plants-12-02910-f003:**
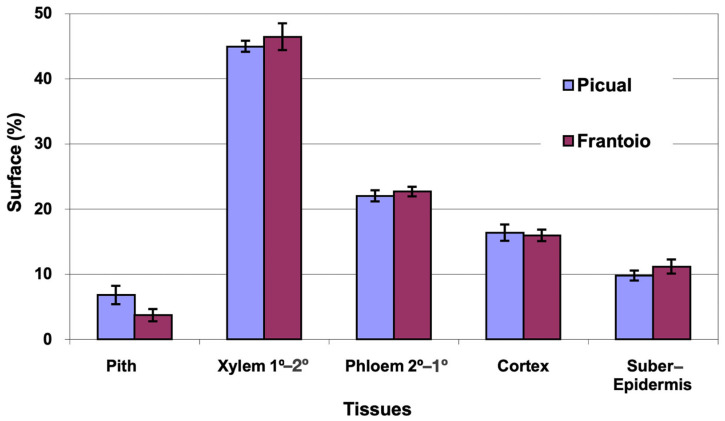
Percentage of the cross section corresponding to different tissues of the apical zones of the main stems of ‘Picual’ and ‘Frantoio’ olive plants. The values represent the mean ± standard error (*n* = 3).

**Figure 4 plants-12-02910-f004:**
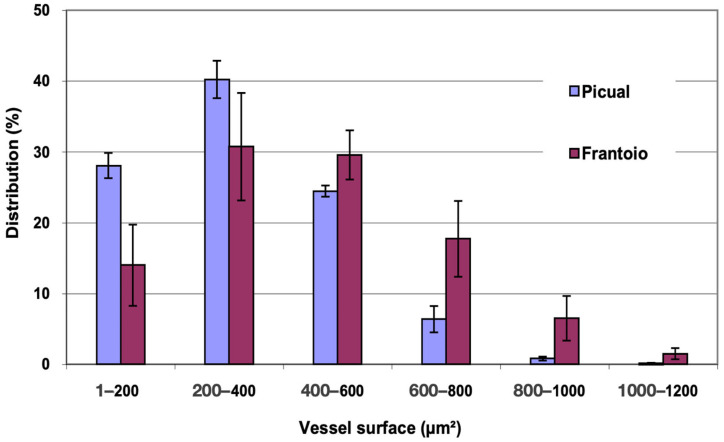
Percentage distribution of the vessels, according to their surface intervals, of the apical zones of the main stems of ‘Picual’ and ‘Frantoio’ olive plants. The values represent the mean ± standard error (*n* = 3).

**Figure 5 plants-12-02910-f005:**
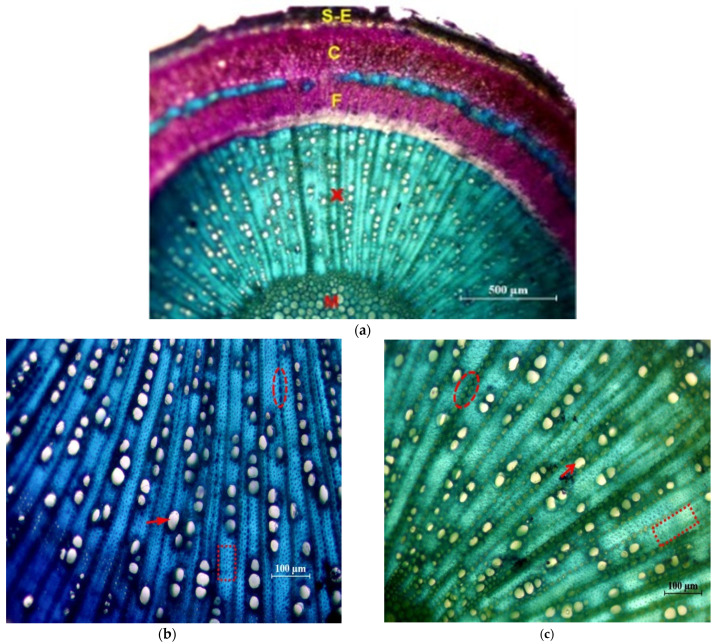
Cross sections of the main stem stained with Toluidine Blue. (**a**) General view of the ‘Picual’ (×4): M = pith; X = 1st and 2nd xylem; F = 2nd and 1st phloem; C = Cortex; S-E = Suber and epidermis. (**b**) Detail of xylem of the ‘Frantoio’ variety (×10): arrow: glass; ellipse: parenchymal radius; inset: fibers. (**c**) Detail of xylem of the ‘Picual’ variety (×10): arrow: glass; ellipse: parenchymal radius; inset: fibers.

**Figure 6 plants-12-02910-f006:**
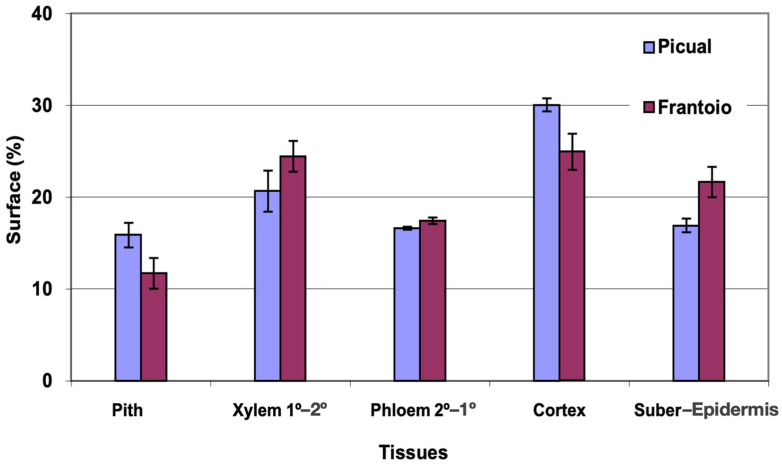
Percentage of the cross section corresponding to different tissues of the lateral shoots of ‘Picual’ and ‘Frantoio’ olive plants. The values represent the mean ± standard error (*n* = 3).

**Figure 7 plants-12-02910-f007:**
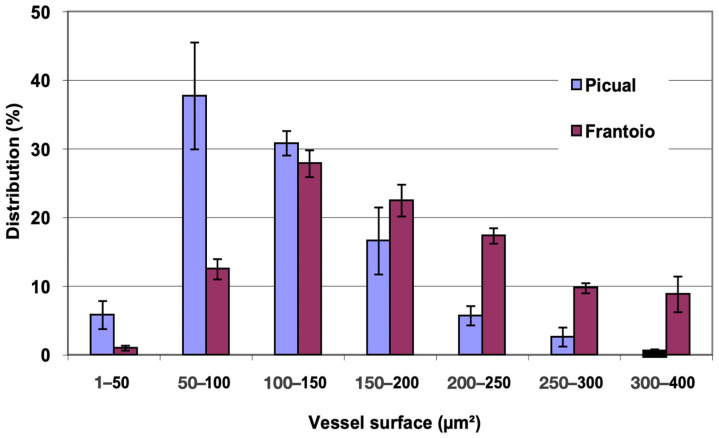
Percentage distribution of the vessels, according to their surface intervals, of lateral shoots of ‘Picual’ and ‘Frantoio’ olive plants. The values represent the mean ± standard error (*n* = 3).

**Figure 8 plants-12-02910-f008:**
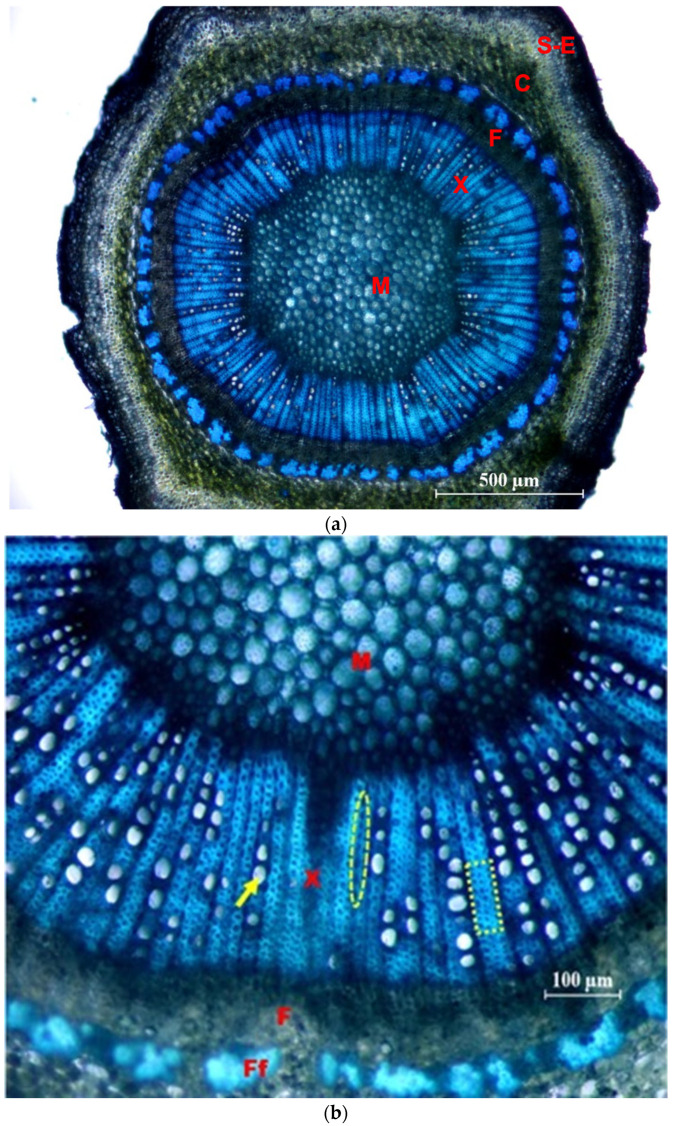
Cross sections of the side shoots of the ‘Frantoio’ variety, stained with Toluidine Blue. (**a**) General view (×4): M = pith; X = 1st and 2nd xylem; F = 2nd and 1st phloem; C = Cortex; S-E = Suber and epidermis. (**b**) Detail (×10): M = pith; X = 1st and 2nd xylem; F = 2nd phloem; Ff = fibers of the 1st phloem; arrow: glass; ellipse: parenchymal radius; inset: fibers.

**Figure 9 plants-12-02910-f009:**
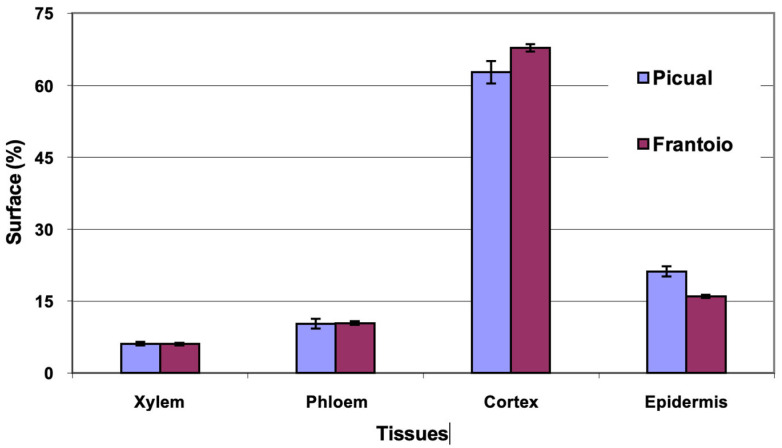
Percentage of the cross section corresponding to different tissues of the leaf petioles of ‘Picual’ and ‘Frantoio’ olive plants. The values represent the mean ± standard error (*n* = 3).

**Figure 10 plants-12-02910-f010:**
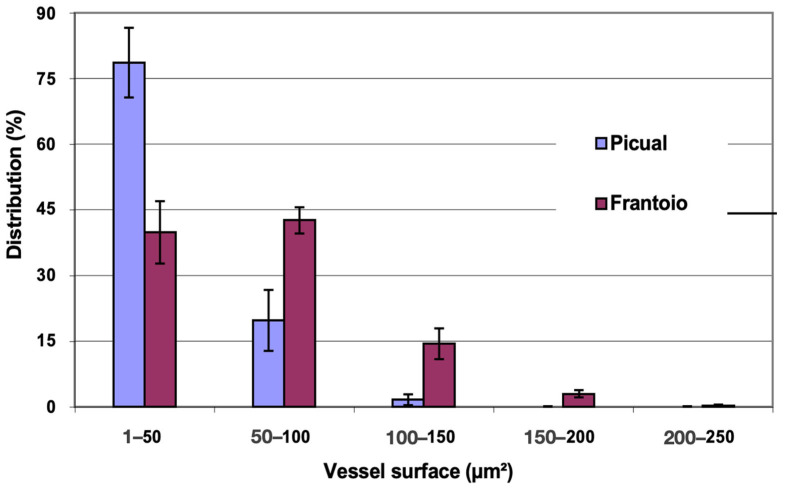
Percentage distribution of the vessels, according to their surface intervals, of the leaf petioles of ‘Picual’ and ‘Frantoio’ olive plants. The values represent the mean ± standard error (*n* = 3).

**Figure 11 plants-12-02910-f011:**
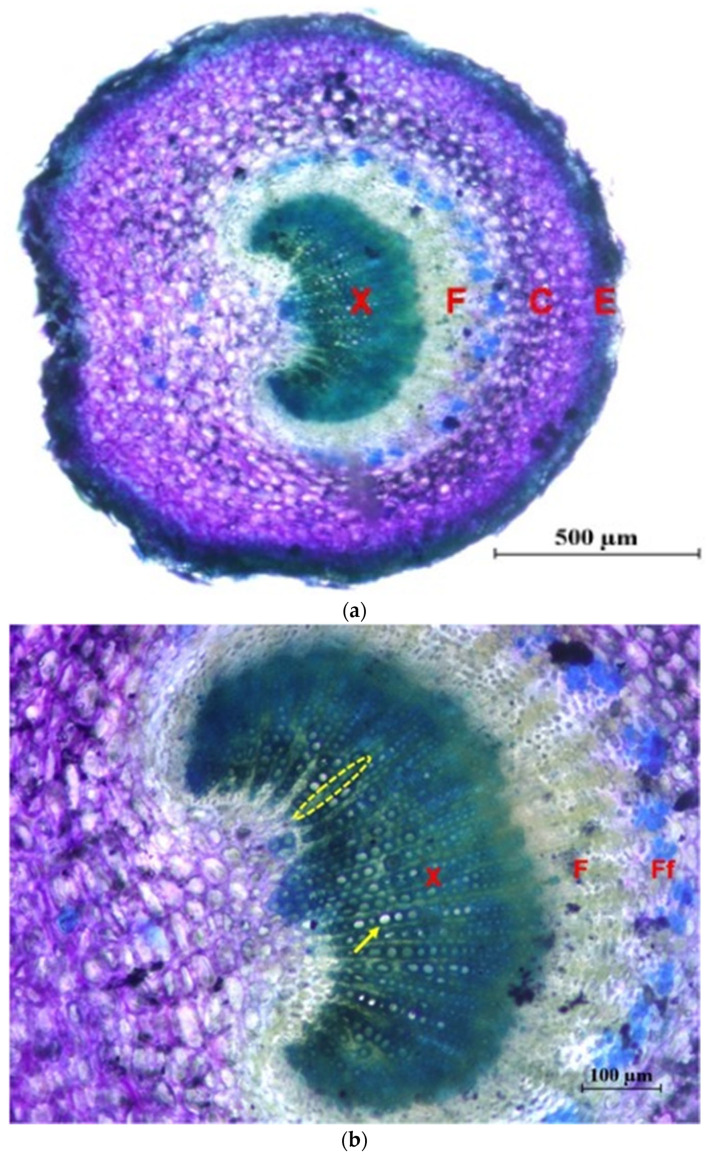
Cross sections of the leaf petiole of the ‘Frantoio’ variety stained with Toluidine Blue. (**a**) General view (×4): X = Xylem; F = Phloem; C = Cortex; E = epidermis. (**b**) Detail (×10): X = xylem; F = phloem; Ff = phloem fibers; arrow: glass; ellipse: parenchymal radius.

**Figure 12 plants-12-02910-f012:**
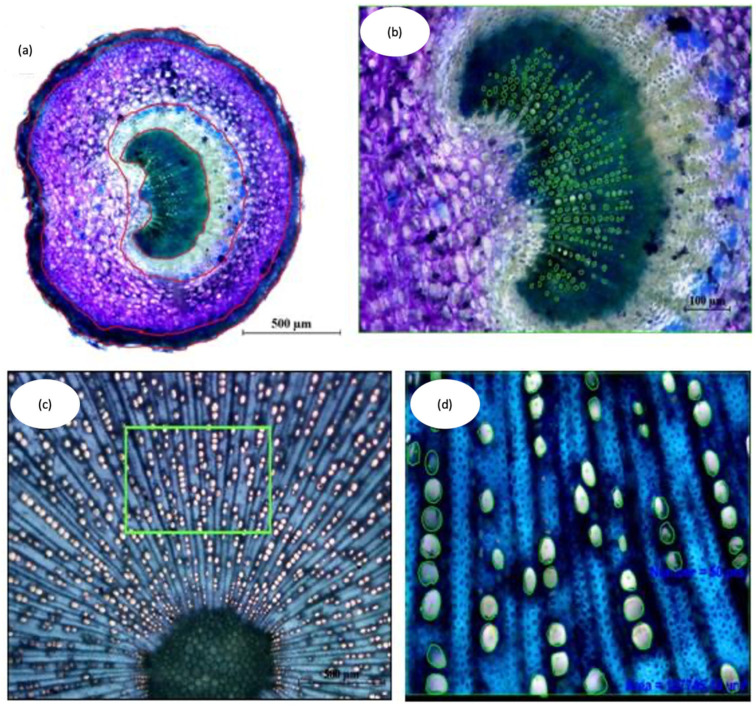
Analysis of the anatomical characteristics of cross section images using the NIS-Elements D 2.3 computer program. (**a**) Measure of the surface of cross section of the petiole (×4); (**b**) Measure of the surface and number of vessels of the Petiole xylem (×10); (**c**) Methodology used to obtain representative images in the xylem of the stem (×4) wherein the box indicates the sector analyzed; (**d**) Measure of the number of vessels of xylem of the main stem (×10).

**Table 1 plants-12-02910-t001:** Quantitative characteristics of the xylems of the basal zones of the main stems of olive plants ^1^.

Olive Variety	Radii Density (Radii/mm)	Vessel Density (Vessels/mm^2^)	Mean Vessel Surface (µm^2^)	TCS ^2^ (%)	RTC ^3^ (μm^4^/μm^2^)
‘Picual’	21.1 ± 1.2 ^a^	164 ± 28 ^a^	453 ± 40 ^a^	7.2 ± 1.1 ^a^	3.9 ± 0.9 ^a^
‘Frantoio’	24.1 ± 0.5 ^a^	248 ± 16 ^b^	527 ± 16 ^a^	12.7 ± 1.0 ^b^	7.9 ± 0.7 ^b^

^1^ Values are the mean (±standard error, *n* = 3). In each column values followed by different letters are significantly different (*p* ≤ 0.05). ^2^ Transverse Conducting Surface: percentage of surface of vessels with respect to the surface of xylem examined. ^3^ Relative Theoretical Conductance: sum of the fourth power of the radii of all the vessels included in the xylem surface examined when divided by said surface.

**Table 2 plants-12-02910-t002:** Quantitative characteristics of the xylems of the apical zones of the main stems of olive plants ^1^.

Olive Variety	Radii Density (Radii/mm)	Vessel Density (Vessels/mm^2^)	Mean Vessel Surface (µm^2^)	TCS ^2^ (%)	RTC ^3^ (μm^4^/μm^2^)
‘Picual’	25 ± 2.4 ^a^	224 ± 7 ^a^	323 ± 12 ^a^	7.2 ± 0.1 ^a^	3.1 ± 0.1 ^a^
‘Frantoio’	26 ± 1.9 ^a^	290 ± 14 ^b^	453 ± 66 ^a^	12.9 ± 1.2 ^b^	7.3 ± 1.5 ^a^

^1^ Values are the mean (±standard error, *n* = 3). In each column, values followed by different letters are significantly different (*p* ≤ 0.05). ^2^ Transverse Conducting Surface: percentage of surface of vessels with respect to the surface of xylem examined. ^3^ Relative Theoretical Conductance: sum of the fourth power of the radii of all the vessels included in the xylem surface examined, divided by said surface.

**Table 3 plants-12-02910-t003:** Quantitative characteristics of the xylems of the lateral shoots of olive plants ^1^.

Olive Variety	Radii Density (Radii/mm)	Vessel Density (Vessels/mm^2^)	Mean Vessel Surface (µm^2^)	TCS ^2^ (%)	RTC ^3^ (μm^4^/μm^2^)
‘Picual’	33.6 ± 0.2 ^a^	519 ± 27 ^a^	119 ± 12 ^a^	6.1 ± 0.5 ^a^	0.9 ± 0.2 ^a^
‘Frantoio’	33.5 ± 0.6 ^a^	588 ± 6 ^a^	181 ± 6 ^a^	10.6 ± 0.4 ^b^	2.3 ± 0.2 ^b^

^1^ Values are the mean (±standard error, *n* = 3). In each column values followed by different letters are significantly different (*p* ≤ 0.05). ^2^ Transverse Conducting Surface: percentage of surface of vessels with respect to the surface of xylem examined. ^3^ Relative Theoretical Conductance: sum of the fourth power of the radii of all the vessels included in the xylem surface examined when divided by said surface.

**Table 4 plants-12-02910-t004:** Quantitative characteristics of the xylems of the apical zones of the petioles of leaves of olive plants ^1^.

Olive Variety	Radii Density (Radii/mm)	Vessel Density (Vessels/mm^2^)	Mean Vessel Surface (µm^2^)	TCS ^2^ (%)	RTC ^3^ (μm^4^/μm^2^)
‘Picual’	40.0 ± 4.4 ^a^	1404 ± 85 ^a^	38 ± 4.3 ^a^	5.4 ± 0.8 ^a^	0.3 ± 0.1 ^a^
‘Frantoio’	40.8 ± 2.1 ^a^	1839 ± 13 ^b^	66 ± 6.2 ^b^	12.2 ± 1.1 ^b^	1.1 ± 0.2 ^b^

^1^ Values are the mean (±standard error, *n* = 3). In each column values followed by different letters are significantly different (*p* ≤ 0.05). ^2^ Transverse Conducting Surface: percentage of surface of vessels with respect to the surface of xylem examined. ^3^ Relative Theoretical Conductance: sum of the fourth power of the radii of all the vessels included in the xylem surface examined when divided by said surface.

## Data Availability

All data analyzed in this study are included in this article.

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
