# Peer review of "Vessel Anatomical Features of ‘Picual’ and ‘Frantoio’, Two Olive Cultivars Different in Resistance against Verticillium Wilt of Olive"

_plants, 2023, doi:10.3390/plants12162910_

Round 1

Reviewer 1 Report

Dear authors my only criticism for your manuscript is the lack of statistics in the figures. This should be corrected as you reach conclusions in the text like in Line 94-95 “The highest percentage of average surface in both varieties is occupied by the 94 xylem (54-59%), followed by the phloem (19-24%).” and Line 118-119 “‘Frantoio’ has a lower 118 percentage of vessels with a smaller surface and a higher percentage of those with a larger 119 surface.” These and all your statements concerning the results coming from the Figures should be supported by statistics. Otherwise, the manuscript presents a number of interesting results about plant resistance against a significant plant pathogen, like it is Verticillium dahliae

Author Response

Dear authors my only criticism for your manuscript is the lack of statistics in the figures. This should be corrected as you reach conclusions in the text like in Line 94-95 “The highest percentage of average surface in both varieties is occupied by the 94 xylem (54-59%), followed by the phloem (19-24%).” and Line 118-119 “‘Frantoio’ has a lower 118 percentage of vessels with a smaller surface and a higher percentage of those with a larger 119 surface.” These and all your statements concerning the results coming from the Figures should be supported by statistics. Otherwise, the manuscript presents a number of interesting results about plant resistance against a significant plant pathogen, like it is Verticillium dahliae.

Response:

It is true that it would be necessary to include statistics in the figures that represent the percentage occupied by the different sections considered and the areas of the different tissues. However, these percentages were only made to inform the reader of the state of said tissues in the selected varieties (for this reason we do not consider it necessary to apply statistics to them). The main objective of the work was to determine the quantitative characteristics of the xylem and for this reason, ANOVA was only performed on the density of the container, percentage of conductive cross-sectional area (TCS), relative theoretical conductance (RTC), ray density and average surface area of the container. This reduced the length of the article and focused even more on the main message that we want to convey with the work.

Since these figures do not have statistics, one option could be to include them as supplementary material, although from our point of view it is much more comfortable for the reader if they are in the main document.

Reviewer 2 Report

This is a superb paper.  The quality of the images is very high. I liked the discussion section where you not only explained your findings and how they agreed or disagreed with others, but also pointed out that differences may occur under infected conditions.  This is important work, thank you for this contribution. There were a few corrections in the citations.  4) Nature 196;

9) italics for latin names

14) italics for latin names

18) misspelled Verticillium

Also, the format was consistent, but most journals only have the first letter with a capital, and all subsequent words are lower case.  Maybe this journal does it differently.

Author Response

This is a superb paper.  The quality of the images is very high. I liked the discussion section where you not only explained your findings and how they agreed or disagreed with others, but also pointed out that differences may occur under infected conditions.  This is important work, thank you for this contribution. There were a few corrections in the citations.  4) Nature 196;9) italics for latin names; 14) italics for latin names; 18) misspelled Verticillium. Also, the format was consistent, but most journals only have the first letter with a capital, and all subsequent words are lower case.  Maybe this journal does it differently.

Response:

Many thanks for the corrections. All of these have been addressed in the revised manuscript.

It is true that most journals only have the first letter with a capital, and all subsequent words are lower case. However, the citations have been entered automatically using the Mendeley program in the style of the Plants journal.

Reviewer 3 Report

It is my opinion that [Plants] Manuscript ID: plants-2470348 entitled "Vessel Anatomical features of 'Picual' and 'Frantoio', two olive 2 different in resistance against Verticillium Wilt of ol-3 ive 4" is suitable for publication in Plants in present form.

The work is scientifically valid, the experimental protocol is well executed and the technical procedures have been carried out correctly. The results are valuable, new, interesting and useful for researchers in the field and beyond, and represent a significant contribution to the field. The contribution to the understanding of the relationship between morphological aspects and mechanisms of varietal resistance to such a serious pathology for the olive tree as Verticillium dahliae is very important.

Author Response

Many thanks for your comment/suggestions, as you indicated the contribution to the understanding of the relationship between morphological aspects and mechanisms of varietal resistance to such a serious pathology for the olive tree as Verticillium dahliae is very important.

Author Response

Vessel Anatomical features of ‘Picual’ and ‘Frantoio’, two olive cultivars different in resistance against Verticillium Wilt of olive

The above manuscript by Antonio Santos-Rufo et al. describes the anatomical variations in two olive varieties that differ in their resistance to the fungal disease Vertcillium wilt. Publications describing the vessel density and their correlation with disease resistance are scanty. Hence, this manuscript warrants publication in the Journal. English is pretty good. The authors followed methods that are internationally accepted. It would be nice if they can address the following concerns:

Response:

We appreciate your feedback, and all proposed concerns have been addressed as follows:

1) If the vessel density increases, how does it impart disease resistance in olive. This question arises since there are conflicting reports in the literature.

Response:

Lines 363-365 explain our hypothesis regarding how such increases in vessel density impart disease resistance in the olive tree. ‘…proposing that an increase in the density of vessels would favor the reduction of the disease since it would help the plant to increase its conductive capacity.’

Do such plants resist to abiotic stresses also?

Response:

No, it does not have to resist abiotic stresses as well. It is known that plant resistance responses are expressed, among other aspects, as resistance to vascular colonization, once the pathogen has accessed the xylem. Therefore, it is the physical effect of obturation in the vessels of the structures of the fungus (conidia).

2) In Figure 8, the labels for S-E (suber and epidermis) are not visible, though it has been mentioned in the legends. Symbols S-E must be visible.

Response:

Thank you for the suggestion. However, In Figure 8 (a) (as indicated in figure 8 (a) caption), the label for S-E (suber and epidermis) is visible.

3) To what percentage level, or extent, the differences in anatomical differences influence the fungal resistance?

Response:

In the case of Frantoio and Picual, more than a 60% difference has been observed in the percentage level, or extension of the disease (López-Escudero et al., 2014).

Do such anatomical changes also help in bacterial infections?

Response:

Yes, it does. Similar to what occurs between olive trees with V. dahliae, a “gum” deposit was observed in the xylem vessels of coffee trees infected with Xylella fastidiosa (Queiroz -Voltan et al., 1998; https://doi.org/10.1590/S0006-87051998000100003). Therefore, it might be possible that the increase in vessels also increases resistance in the case of diseases caused by bacteria.